# Spectro-Image Analysis with Vision Graph Neural Networks and Contrastive Learning for Parkinson’s Disease Detection

**DOI:** 10.3390/jimaging11070220

**Published:** 2025-07-02

**Authors:** Nuwan Madusanka, Hadi Sedigh Malekroodi, H. M. K. K. M. B. Herath, Chaminda Hewage, Myunggi Yi, Byeong-Il Lee

**Affiliations:** 1Digital Healthcare Research Center, Pukyong National University, Busan 48513, Republic of Korea; nuwanv@pknu.ac.kr (N.M.); myunggi@pknu.ac.kr (M.Y.); 2Industry 4.0 Convergence Bionics Engineering, Pukyoung National University, Busan 48513, Republic of Korea; hadi_sedigh@pukyong.ac.kr (H.S.M.); kasunkh@pukyong.ac.kr (H.M.K.K.M.B.H.); 3Cardiff School of Technologies, Cardiff Metropolitan University, Cardiff CF23 6PS, UK; chewage@cardiffmet.ac.uk; 4Division of Smart Healthcare, College of Information Technology and Convergence, Pukyong National University, Busan 48513, Republic of Korea

**Keywords:** Vision Graph Neural Networks, supervised contrastive learning, Parkinson’s disease, speech analysis, frequency band decomposition, spectro-temporal analysis

## Abstract

This study presents a novel framework that integrates Vision Graph Neural Networks (ViGs) with supervised contrastive learning for enhanced spectro-temporal image analysis of speech signals in Parkinson’s disease (PD) detection. The approach introduces a frequency band decomposition strategy that transforms raw audio into three complementary spectral representations, capturing distinct PD-specific characteristics across low-frequency (0–2 kHz), mid-frequency (2–6 kHz), and high-frequency (6 kHz+) bands. The framework processes mel multi-band spectro-temporal representations through a ViG architecture that models complex graph-based relationships between spectral and temporal components, trained using a supervised contrastive objective that learns discriminative representations distinguishing PD-affected from healthy speech patterns. Comprehensive experimental validation on multi-institutional datasets from Italy, Colombia, and Spain demonstrates that the proposed ViG-contrastive framework achieves superior classification performance, with the ViG-M-GELU architecture achieving 91.78% test accuracy. The integration of graph neural networks with contrastive learning enables effective learning from limited labeled data while capturing complex spectro-temporal relationships that traditional Convolution Neural Network (CNN) approaches miss, representing a promising direction for developing more accurate and clinically viable speech-based diagnostic tools for PD.

## 1. Introduction

Parkinson’s disease (PD) affects over six million people worldwide, representing the second most common neurodegenerative disorder, with projections indicating a doubling of cases by 2030 due to global population aging [1,2]. Early detection remains a critical challenge, as current diagnostic approaches rely heavily on clinical observation of motor symptoms that typically appear only after 50–70% of dopaminergic neurons have already been lost [3]. This diagnostic delay significantly impacts treatment effectiveness and patient outcomes, creating an urgent need for objective, accessible screening methods that can detect the disease in its earliest stages.

Speech and voice impairments emerge as promising biomarkers for early PD detection, affecting more than 90% of patients and often preceding classic motor symptoms by several years [4,5,6]. The underlying pathophysiology involves progressive dysfunction of the respiratory, phonatory, and articulatory systems, resulting in characteristic patterns of hypokinetic dysarthria including reduced vocal loudness, monotone speech, and imprecise articulation [7]. These changes can be quantitatively analyzed through digital signal processing, offering potential for automated, non-invasive diagnostic tools.

The field of speech-based PD detection has evolved significantly over the past decade. Early approaches focused on extracting handcrafted acoustic features such as fundamental frequency variations, jitter, shimmer, and mel-frequency cepstral coefficients (MFCCs), combined with traditional machine learning classifiers [4,5]. Recent studies have achieved over 90% accuracy in distinguishing PD patients from healthy controls (HC) using sustained vowel recordings [8,9,10]. Contemporary approaches have further advanced the field, with wavelet scattering networks showing promising early detection capabilities [11], machine learning-assisted speech analysis demonstrating improved diagnostic accuracy [12], ensemble methods achieving robust classification performance [13], and telephone-collected voice recordings proving reliable for PD identification [14]. A recent systematic review highlighted both the potential and limitations of current ML approaches for PD detection [15]. However, these methods were inherently limited by their reliance on predetermined feature sets and inability to capture complex, non-linear patterns in speech data.

The emergence of deep learning has transformed this landscape, with convolutional neural networks (CNNs) and hybrid CNN-LSTM architectures achieving remarkable performance improvements [16,17]. Recent studies report accuracies exceeding 90% through end-to-end learning approaches that automatically extract discriminative features from spectro-temporal representations of speech signals [18,19]. This paradigm shift toward treating audio as spectro-images has enabled the application of advanced computer vision techniques to speech analysis, with mel spectrograms emerging as the preferred representation due to their perceptually-motivated frequency scaling.

Despite these advances, several controversial and diverging perspectives persist in the field. A significant methodological controversy involves validation strategies, with many studies employing inappropriate cross-validation schemes that lead to overly optimistic performance estimates and poor generalization to new populations.

Additionally, there remains substantial disagreement regarding optimal speech tasks, with some researchers advocating for sustained phonations while others argue for connected speech or reading tasks [18,19]. Language dependency represents another contentious issue, as models trained on one language often fail to generalize across linguistic boundaries, limiting clinical applicability.

Current approaches also face fundamental limitations in their architectural assumptions. Most existing methods employ traditional CNNs that process spectro-images as regular images, potentially overlooking the complex relational structures inherent in spectro-temporal data. The spatial relationships between different frequency bands and temporal segments in spectrograms may be better captured through approaches that explicitly model these connections as graph structures. Furthermore, the chronic shortage of labeled medical data continues to constrain model development, with traditional supervised learning requiring extensive expert-annotated datasets that are expensive and time-consuming to collect.

Graph neural networks (GNNs) have demonstrated remarkable success in modeling complex relational structures across various domains, offering potential solutions to these limitations [20,21,22]. Vision Graph Neural Networks (ViGs) represent a recent innovation that treats images as graphs of visual elements, enabling the modeling of long-range dependencies and complex spatial relationships [23]. Recent comprehensive reviews have highlighted the growing success of GNNs in medical image analysis and disease diagnosis, demonstrating their ability to capture irregular spatial patterns and complex relationships that traditional CNNs often miss [24,25]. Concurrently, contrastive learning has emerged as a powerful self-supervised approach that addresses data scarcity by learning representations from unlabeled data, achieving significant improvements in data efficiency and model robustness [26,27]. Contrastive learning has shown particular promise in medical audio analysis, with recent studies demonstrating its effectiveness for respiratory sound classification [28], medical time series analysis [29], and general-purpose audio representations [30,31].

The selection of ViGs for this application is motivated by three key limitations in current approaches: First, traditional CNNs process spectrograms as regular images, failing to capture the complex relational dependencies between frequency bands and temporal segments that are crucial for detecting subtle PD-specific speech characteristics. Second, the spatial relationships in spectro-temporal data are inherently graph-like, where frequency bins and time frames exhibit long-range dependencies that standard convolution operations cannot effectively model. Third, the adoption of vision-based models for speech analysis represents a paradigm shift that treats audio signals as spectro-images, enabling the application of advanced computer vision techniques to capture dysarthric speech patterns that manifest as complex spectro-temporal relationships across different frequency bands, which are better modeled through graph structures than conventional grid-based representations.

This study addresses the fundamental research question: Can ViGs combined with supervised contrastive learning effectively model the complex spectro-temporal relationships in speech signals to achieve superior PD detection compared to traditional CNN-based approaches? Specifically, we investigate whether explicitly modeling spectrograms as graphs rather than regular images enables better capture of the subtle, distributed patterns characteristic of dysarthric speech.

To answer this research question, the main aim of this work is to address current limitations in speech-based PD detection by developing a novel framework that integrates ViGs with supervised contrastive learning for enhanced spectro-temporal image analysis. Our approach introduces a frequency band decomposition strategy that transforms raw audio into three complementary spectral representations (low: 0–2 kHz, mid: 2–6 kHz, high: 6 kHz+), capturing distinct PD-specific characteristics across the frequency spectrum. The framework processes mel multi-band spectro-temporal representations through a ViG architecture that models complex graph-based relationships between spectral and temporal components, trained using a supervised contrastive objective that learns discriminative representations distinguishing PD-affected from healthy speech patterns.

The principal conclusions of this study demonstrate that the proposed ViG-contrastive framework achieves superior classification performance compared to existing approaches, with significant improvements in both accuracy and robustness. The integration of graph neural networks with contrastive learning enables effective learning from limited labeled data while capturing complex spectro-temporal relationships that traditional CNN approaches miss. These findings suggest that explicitly modeling the relational structure of spectro-images, combined with advanced self-supervised learning techniques, represents a promising direction for developing more accurate and clinically viable speech-based diagnostic tools for PD and potentially other neurological conditions affecting speech production.

## 2. Materials and Methods

### 2.1. Datasets and Data Availability

This study utilized three internationally recognized datasets from Italy, Colombia, and Spain to create a comprehensive multi-institutional corpus, enhancing robustness and cross-site generalizability of PD detection models. The strategic combination provides unprecedented diversity in recording environments, equipment specifications, demographic distributions, and clinical protocols while maintaining phonetic consistency through sustained vowel productions (/a/, /e/, /i/, /o/, /u/) across Italian, Spanish, and Colombian Spanish. Following comprehensive preprocessing and systematic vowel extraction procedures, the consolidated corpus comprised 3530 sustained vowel recordings (1834 PD, 1696 healthy controls) obtained from 258 participants (131 individuals with PD, 127 neurologically healthy controls) distributed across the three institutional contexts.

#### 2.1.1. Dataset Characteristics

The multi-institutional approach represents a significant methodological advancement in cross-institutional PD detection research. Table 1 presents the comprehensive characteristics of each dataset. Each dataset contains extensive speech tasks, including connected speech, reading passages, diadochokinetic exercises, and spontaneous speech, demonstrating the comprehensive nature of these corpora for voice disorder research. However, this study utilized only sustained vowel phonations to ensure phonetic consistency and optimal spectro-temporal analysis across different institutional contexts.

#### 2.1.2. Individual Dataset Descriptions

The ItalianPVS corpus, as described by Dimauro et al. [32] and subsequently validated by Dimauro et al. [33], is publicly accessible through the IEEE DataPort repository (https://ieee-dataport.org/open-access/italian-parkinsons-voice-and-speech) accessed on 22 January 2025.

This carefully curated dataset encompasses voice recordings from a cohort of 50 native Italian speakers, comprising 22 neurologically healthy control participants (12 female, 10 male) and 28 individuals diagnosed with idiopathic PD (9 female, 19 male). The participant groups were rigorously age-matched, with control subjects exhibiting a mean age of 67.1 ± 5.2 years and PD patients demonstrating a mean age of 67.2 ± 4.8 years. All recordings were acquired at the Department of Computer Science, University of Bari, Italy, under controlled laboratory conditions using professional-grade audio equipment and standardized elicitation protocols established for voice research.

The PC-GITA (Parkinson’s Disease-Group of Intelligent Technologies for Assessment) corpus, as originally developed and described by Orozco-Arroyave et al. [34], represents a meticulously balanced speech database accessible through Universidad de Antioquia, Medellín, Colombia, upon academic request. This corpus encompasses comprehensive speech recordings from 100 native Colombian Spanish speakers, with equal stratification between 50 individuals diagnosed with idiopathic PD and 50 neurologically healthy control subjects. Demographic matching was rigorously maintained across multiple variables, with age distributions spanning 31 to 86 years and perfect gender balance (25 males and 25 females per group) as detailed in the original corpus documentation. All audio acquisitions were conducted within acoustically controlled environments at Clínica Noel, Medellín, Colombia, utilizing professional-grade dynamic omnidirectional microphones with high-fidelity recording specifications (44.1 kHz sampling frequency, 16-bit resolution).

The Neurovoz corpus, as comprehensively documented by Moro-Velázquez et al. [35] and subsequently detailed in their Scientific Data publication [36], constitutes the most extensive publicly available Castilian Spanish speech database specifically designed for Parkinsonian speech research. This corpus is accessible through Universidad Politécnica de Madrid following established data sharing agreements for academic research purposes. The dataset encompasses recordings from 108 native Castilian Spanish speakers, comprising 53 individuals with clinically confirmed idiopathic PD and 55 neurologically healthy control participants, with careful demographic matching across age and gender variables as described in the original cohort characterization. The extensive data collection, conducted between 2015 and 2017 through collaborative efforts between the Bioengineering and Optoelectronics Group of Universidad Politécnica de Madrid and the Departments of Otorhinolaryngology and Neurology at Hospital General Universitario Gregorio Marañón, yielded 2903 individual audio recordings with a mean of 26.88 ± 3.35 recordings per participant.

#### 2.1.3. Multi-Institutional Dataset Integration and Validation

While Table 1 presents the basic characteristics of our three datasets, the strategic combination of these internationally recognized corpora requires additional justification to address potential methodological concerns inherent in multi-institutional speech analysis.

Phonetic Compatibility Across Romance Languages. All three datasets focus on sustained vowel phonations (/a/, /e/, /i/, /o/, /u/) across Romance languages (Italian, Colombian Spanish, Castilian Spanish). Romance languages share common Latin origins with similar vowel systems (/a/, /e/, /i/, /o/, /u/), comparable articulatory patterns for sustained vowels, and similar laryngeal control mechanisms affected by PD pathophysiology. This linguistic compatibility enables reliable cross-cultural comparison while maintaining the phonetic consistency essential for PD detection. The established literature supports cross-Romance language speech analysis for neurological disease detection, demonstrating that fundamental voice quality parameters remain consistent across these linguistic variants [4,6,35].

The three datasets provide complementary demographic coverage, addressing limitations of single-institution studies. Age diversity spans focused cohorts (Italian PVS ~67 years) to broad ranges (PC-GITA 31–86 years) and age-matched adult populations (Neurovoz). Geographic diversity encompasses Southern Europe (Italy), South America (Colombia), and Western Europe (Spain), representing different healthcare systems and diagnostic protocols. The combined sample size of 258 participants substantially exceeds individual datasets (50–108 participants), providing enhanced statistical power essential for deep learning approaches. Multi-institutional combination addresses key limitations identified in recent systematic reviews [15], including reduction of single-institution bias and overfitting to specific recording conditions, enhanced model generalizability across different clinical environments, larger sample sizes essential for robust deep learning model training, and cross-linguistic validation of PD speech biomarkers.

### 2.2. Data Preprocessing and Feature Engineering

#### 2.2.1. Audio Preprocessing Pipeline

All audio recordings underwent standardized preprocessing to ensure temporal and spectral consistency across datasets, as shown in Figure 1. The preprocessing pipeline was designed to address the inherent variability in recording conditions, equipment specifications, and signal characteristics across the three institutional datasets while preserving the speech biomarkers essential for accurate PD detection.

To address recording heterogeneity across institutional contexts, we implemented standardized preprocessing procedures designed for multi-dataset integration. Audio resampling harmonized all datasets to 16 kHz (from original 44.1 kHz PC-GITA and 16 kHz for Italian PVS and Neurovoz) using librosa’s high-quality resampling algorithm [37] with anti-aliasing filters to maintain signal integrity while ensuring computational efficiency suitable for clinical deployment. Dynamic range normalization using z-score standardization was applied per dataset before combination to address equipment-specific amplitude variations across institutions.

Raw audio signals *x*(*t*) were first normalized in amplitude to prevent dynamic range variations from affecting subsequent analysis:(1)$$x_{norm}(t)=\frac{x\left(t\right)-\mu_{x}}{\sigma_{x}}$$
where $\mu_{x}$ and $\sigma_{x}$ represent the signal mean and standard deviation, respectively. Silent regions were automatically detected using energy-based voice activity detection with a threshold of −40 dB relative to peak amplitude and subsequently removed to focus analysis on voiced segments containing PD-related information. A minimum duration threshold of 2.4 s was enforced, while recordings were segmented to a standardized duration of 2.4 s to maintain computational efficiency.

Data augmentation techniques were applied to generate robust training samples (Figure 1b), including pitch shifting, time stretching, and additive noise [38,39] while preserving speech characteristics essential for voice disorder detection. Quality control measures ensured that augmentation parameters remained within physiologically plausible ranges to maintain diagnostic validity.

The frequency band decomposition approach transforms the preprocessed audio signals (Figure 1a) into three distinct spectro-temporal representations corresponding to different aspects of dysarthric speech characteristics, grounded in the pathophysiology of Parkinsonian speech disorders, which manifest differently across the frequency spectrum. Voice tremor and fundamental frequency instability primarily affect lower frequencies (0–2 kHz), formant frequency modifications and articulation deficits impact mid-frequencies (2–6 kHz), while breathiness and noise characteristics are most prominent at higher frequencies (6 kHz and above) [4,6,7].

For each audio segment *x*(*t*), the Short-Time Fourier Transform (STFT) is computed using optimal parameters for speech analysis:(2)$$X\left(m,\;k\right)=\sum_{n=0}^{N-1}x(n)\omega (n-mH)e^{-j2\pi kn/N}$$
where *N* = 2048 samples (128 ms window at 16 kHz), *H* = 108 samples (6.75 ms hop), and $\omega \left(n\right)\;$represents a Hann window function optimized for speech signal analysis.

The mel-scale transformation better approximates human auditory perception and enhances the representation of perceptually relevant features [40]. The mel-scale conversion is implemented as:(3)$$mel\left(f\right)=2595\;{log}_{10}\left(1+\frac{f}{700}\right)$$

For each frequency band *b*, band-limited mel-spectrograms are computed using triangular filter banks:(4)$$S_{b}\left(m,\;k\right)=\sum_{n=f_{min}^{b}}^{f_{max}^{b}}{\left|X\left(m,\;n\right)\right|}^{2}\cdot M_{k}(n)$$
where $M_{k}(n)$ represents the *k*-th mel filter coefficient, and frequency ranges. The resulting spectrograms undergo logarithmic compression to reduce dynamic range and improve numerical stability:(5)$$L_{b}\left(m,\;k\right)=10{log}_{10}\left(\frac{S_{b}\left(m,\;k\right)}{max{(S}_{b})}+\epsilon \right)$$
where $\epsilon =10^{-10}$ prevents numerical instability for zero-valued bins.

Independent min-max normalization is applied to each frequency band to ensure balanced contribution to the final representation:(6)$${\hat{L}}_{b}\left(m,\;k\right)=\frac{L_{b}\left(m,\;k\right)-min\left(L_{b}\right)}{max\left(L_{b}\right)-min\left(L_{b}\right)+\epsilon }$$

The normalized spectrograms are resized to 224 × 224 pixels using bilinear interpolation to match standard computer vision input dimensions:(7)$${\hat{L}}_{b}^{224}\left(i,\;j\right)=BilinearInterp\left({\hat{L}}_{b},\frac{i\cdot M}{224},\;\frac{j\cdot K}{224}\right)$$
where *M* and *K* represent the original spectrogram dimensions (typically 256 mel bins × 224-time frames).

The final three-channel feature tensor is constructed as:(8)$$F=\left[\begin{array}{c} {\hat{L}}_{low}^{224}\\ {\hat{L}}_{mid}^{224}\\ {\hat{L}}_{high}^{224} \end{array}\right]\in R^{3\times 224\times 224}$$

This produces standardized 224 × 224 × 3-pixel feature tensors where each channel represents a specific frequency band (Red = Low Freq 0–2 kHz, Green = Mid Freq 2–6 kHz, Blue = High Freq 6 kHz+), creating RGB-like representations optimized for vision-based neural network processing (Figure 1f).

#### 2.2.2. Multi-Scale Data Augmentation Framework

The data augmentation pipeline was specifically designed to enhance model robustness while preserving the speech characteristics essential for PD detection. Data augmentation has become increasingly important in medical machine learning applications, with systematic reviews demonstrating its critical role in addressing data scarcity challenges [41]. For dysarthric speech specifically, two-stage augmentation strategies have shown particular promise in improving ASR performance [42]. The augmentation transformations were applied probabilistically during training to introduce beneficial invariances without compromising diagnostic information.

The selected augmentation parameters were specifically designed to simulate natural variations within physiologically plausible ranges while preserving PD-specific characteristics. Each augmentation technique is grounded in the known pathophysiology of Parkinsonian speech disorders to ensure clinical validity rather than arbitrary regularization.

Pitch shifting (±2 semitones) mimics the fundamental frequency variations characteristic of PD patients due to reduced laryngeal control and vocal fold rigidity [6,7]. Studies have documented that PD patients exhibit irregular fundamental frequency patterns and reduced pitch variability, making pitch augmentation clinically relevant for capturing these natural variations [4,5]. Time stretching (0.9–1.1×) reflects the speaking rate variations characteristic of hypokinetic dysarthria, where PD patients often demonstrate either accelerated speech (festinating speech) or bradykinesia-induced slow speech patterns. This augmentation simulates the temporal irregularities commonly observed in PD speech production without altering spectral content [7,43]. Volume adjustments (0.6–1.4×) simulate the reduced vocal loudness (hypophonia) that affects over 90% of PD patients, as well as variations in recording conditions across clinical environments. Hypophonia represents one of the earliest and most consistent speech manifestations in PD, making volume variation augmentation both physiologically relevant and methodologically necessary. Additive white noise (0.003–0.02 noise level) models the increased breathiness and aspiration noise commonly present in PD speech due to incomplete vocal fold closure and reduced respiratory support. This augmentation also accounts for environmental recording variations across different clinical settings [44]. Time shifting (±20% duration) preserves speech content while introducing temporal variability that accounts for natural variations in speech onset timing and the temporal coordination difficulties often observed in PD patients during speech initiation [45].

These transformations introduce beneficial invariances that improve model robustness while maintaining the integrity of dysarthric speech patterns essential for accurate PD detection. The parameter ranges were constrained to remain within physiologically observed variations in PD speech to ensure augmented samples remain clinically valid representations of potential disease manifestations.

Temporal domain augmentations include time shifting implemented as a circular shift operation within a constrained range to preserve speech content while introducing temporal variability, formulated as:(9)$$x_{shifted}\left(t\right)=x\left(\left(t+\Delta t\right)mod\;T\right)$$
where Δ*t* ~ *U*(−0.2*T*, 0.2*T*) represents a uniformly distributed time offset.

Random volume adjustment was applied multiplicatively to simulate variations in recording conditions according to:(10)$$x_{volume}\left(t\right)=\alpha \cdot x\left(t\right),\;\;\alpha \;\sim \;u(0.6,\;1.4)$$

Additive white noise was introduced to improve robustness against recording artifacts using:(11)$$x_{noise}\left(t\right)=x\left(t\right)+\eta (t),\;\;\eta (t)\;\sim \;Ν(0,\;\sigma^{2})$$

Spectral domain augmentations were designed to simulate natural variations in speech production while maintaining the integrity of speech characteristics. Pitch shifting was applied in the frequency domain to simulate natural variations in fundamental frequency while preserving formant relationships, implemented as:(12)$$X_{pitch}\left(f\right)=X\left(f\cdot 2^{n/12}\right)$$
where *n* ~ *U*(−2, 2) represents the pitch shift in semitones.

Time stretching modified the temporal characteristics while maintaining spectral content, enabling the model to become invariant to speaking rate variations while preserving spectral and articulatory characteristics.(13)$$X_{strech}\left(t\right)=x\left(t\cdot r\right),\;\;r\;\sim \;U(0.9,1.1)$$

### 2.3. Proposed Architecture

#### 2.3.1. Vision Graph Neural Network Architecture

The proposed system employs a ViG architecture adapted for audio classification through spectro-temporal analysis. ViGs are chosen for their superior capability to model complex spatial relationships and long-range dependencies in image-like data through dynamic graph construction [23,46], which has shown success in medical image analysis applications [47]. The ViG architecture implementation in this study used random weight initialization following Xavier/Glorot initialization rather than traditional ImageNet pre-training. While conventional CNN approaches commonly leverage ImageNet pre-trained weights, our approach employs a two-stage training strategy: (1) supervised contrastive pre-training on speech spectrograms, followed by (2) joint contrastive classification fine-tuning. This domain-specific pre-training approach avoids potential domain mismatch between natural images and audio spectrograms while enabling the model to learn robust, discriminative features through multi-view contrastive learning. This ensures that the learned graph-based representations are optimally adapted to the spectro-temporal characteristics of speech signals rather than being biased toward natural image features.

The ViG architecture consists of three main components optimized for graph-based spectro-temporal analysis, as illustrated in Figure 2. The graph construction stage converts input spectro-temporal images into graph representations where pixels become nodes and spatial relationships are encoded as edges. The graph processing module applies graph convolution operations to capture complex dependencies between frequency bands and temporal segments, adaptively learning the importance of different spatial relationships to focus on diagnostically relevant patterns. The feature transformation stage employs fully connected layers to convert graph-processed representations into discriminative features for final classification.

The core innovation lies in explicitly modeling spectro-temporal relationships as graphs rather than treating spectrograms as conventional images, enabling the capture of complex dependencies essential for detecting speech patterns. The entire pipeline is trained end-to-end to optimize both graph-based feature extraction and classification performance.

#### 2.3.2. System Overview and Design Principles

The framework operates through two distinct but complementary phases, each serving specific objectives in the learning process. This architectural innovation addresses the fundamental challenges in medical audio classification by combining robust representation learning with task-specific optimization while maintaining computational efficiency for clinical deployment. The two-phase approach enables the model to first establish strong feature representations through contrastive learning before adapting these representations for the specific classification task, as illustrated in Figure 3.

Phase 1 focuses exclusively on learning discriminative feature representations through supervised contrastive learning, establishing a strong foundation for speech pattern recognition. This phase enables the model to learn robust representations that capture the essential characteristics distinguishing PD-affected from healthy speech patterns without the potential interference of classification-specific biases. The supervised contrastive learning component addresses the chronic limitation of small medical datasets by learning from unlabeled augmented views while leveraging diagnostic label information to ensure clinically relevant feature separation.

Phase 2 introduces a joint learning paradigm that maintains the benefits of contrastive representation learning while incorporating explicit classification objectives for PD detection. This sophisticated joint learning paradigm preserves the robust feature representations learned during contrastive pre-training while adapting to the specific requirements of PD classification. The pre-trained ViG backbone serves as the foundation, with its learned representations being simultaneously processed through two parallel pathways designed to optimize both representation quality and classification performance.

### 2.4. Training Methodology

#### 2.4.1. Supervised Contrastive Learning Framework

Medical audio classification presents unique challenges, including limited labeled data, significant class imbalance, and high intra-class variability due to patient-specific dysarthric speech manifestations. Traditional supervised learning approaches often struggle with these constraints, leading to overfitting and poor generalization. To address these limitations, we introduce a supervised contrastive learning framework [26] that learns robust feature representations by maximizing agreement between augmented views of samples from the same class while promoting clear separation between different classes.

The supervised contrastive learning paradigm operates on the principle that semantically similar samples (same diagnostic class) should cluster together in the learned representation space, while dissimilar samples (different classes) should be well-separated. By incorporating explicit label information into the contrastive objective, our approach provides stronger learning signals compared to self-supervised alternatives, leading to more discriminative representations for dysarthric speech patterns analysis. This approach is particularly beneficial for medical applications where the subtle differences between PD-affected speech patterns and healthy patterns require enhanced discriminative capability.

For each audio sample $x_{i}$ with corresponding label $y_{i}$, the framework generates N=2 augmented views through a carefully designed stochastic augmentation pipeline. The augmentation strategy balances two critical objectives: introducing sufficient variation to improve model robustness while preserving the dysarthric speech characteristics essential for accurate PD detection.

#### 2.4.2. Supervised Contrastive Loss Formulation

The supervised contrastive loss extends standard contrastive learning by leveraging label information to define positive and negative pairs explicitly, ensuring that the learned representations reflect clinically meaningful similarities and differences. For a mini-batch B containing 2B augmented views (2 views per sample), the loss is formulated as:(14)$$L_{SupCOn}=\sum_{i\in I}\frac{-1}{\left|P(i)\right|}\sum_{p\in P(i)}log\frac{exp(z_{i}\cdot z_{p}/\tau )}{\sum_{a\in A(i)}exp(z_{i}\cdot z_{a}/\tau )}$$
where: $I=\left\{1,\;2,\;\ldots ,\;2B\right\}$ represents all anchor indices.$P\left(i\right)=\left\{p\in A\left(i\right):\;y_{p}=y_{i}\right\}\;$defines positive samples sharing the same label.$A\left(i\right)=I\backslash \left\{i\right\}\;$includes all samples except the anchor.τ is the temperature parameter controlling distribution concentration.

#### 2.4.3. Two-Phase Training Protocol

The training procedure employs a carefully designed two-phase approach to maximize the benefits of representation learning through contrastive objectives. This methodology enables the model to first establish robust feature representations before adapting them for the specific classification task, ensuring that the learned representations capture the essential PD-specific features while maintaining sufficient flexibility for task-specific adaptation.

Phase 1 focuses exclusively on learning robust feature representations through supervised contrastive loss optimization, establishing a strong foundation for subsequent classification fine-tuning. The optimization objective for this phase is formulated as:(15)$$\theta^{*},\phi^{*}=arg\;{min}_{\theta ,\phi }E_{\left(x,\;y\right)∼D_{train}}\left[L_{SupCon}\left(h_{\phi }\left(f_{\theta }\left(x\right)\right),\;y\right)\right]$$
where θ  represents feature extractor parameters and ϕ  denotes projection head parameters.

Phase 2 introduces the classification objective while maintaining the contrastive learning framework, creating a multi-task learning scenario that balances representation quality with task-specific performance. This joint training approach leverages the robust representations learned during pre-training while adapting them for the specific classification task. The combined loss function is formulated as:(16)$$L_{total}=\lambda_{c}L_{SupCon}+\lambda_{cls}L_{CE}$$
where $\lambda_{c}$ and $\lambda_{cls}$ represent loss weighting hyperparameters, and the classification loss incorporates class balancing to address the inherent imbalance in medical datasets. This dual-pathway design ensures that the model maintains the robust feature representations learned during contrastive pre-training while adapting to the specific nuances of dysarthric speech patterns classification.

## 3. Experimental Setup and Implementation

### 3.1. Dataset Preparation and Augmentation

Following the frequency band decomposition preprocessing described in the previous section, the final experimental dataset comprised 7356 audio segments representing a substantial expansion from the original corpus through systematic segmentation and augmentation procedures. The dataset distribution was carefully designed to maintain diagnostic validity while providing sufficient training examples for robust model development across different data splits, as shown in Table 2. This multi-institutional approach addresses key limitations identified in recent systematic reviews, where many PD detection studies suffer from limited generalizability due to single-institution datasets [15].

The training set received comprehensive augmentation, including both the multi-view generation for contrastive learning and the temporal and spectral domain augmentations described previously. The validation and test sets were kept pristine without augmentation to ensure unbiased evaluation of model performance and generalization capability. The class balance ratios across splits demonstrate reasonable distribution, with the slight imbalance reflecting the natural prevalence patterns observed in clinical populations while remaining within acceptable ranges for robust model training.

Our experimental design addresses potential dataset heterogeneity through several mechanisms specifically designed for cross-institutional data integration. Stratified splitting maintains proportional representation from all three datasets (Italian PVS, PC-GITA, Neurovoz) across training, validation, and test sets as shown in Table 2, ensuring that model evaluation reflects performance across diverse institutional contexts rather than single-source validation. Consistent evaluation metrics are applied across all institutional contexts, with augmentation applied exclusively to the training set while keeping validation and test sets pristine for unbiased evaluation. This approach enables assessment of cross-institutional generalizability while maintaining the methodological rigor essential for medical AI applications.

### 3.2. Hyperparameter Optimization

Systematic hyperparameter optimization was conducted for the supervised contrastive learning framework using a grid search approach across critical parameters that significantly influence contrastive representation learning quality. Given the computational complexity of training multiple ViG architectures, we performed this optimization using the largest ViG-S-GELU model to ensure the selected hyperparameters would be effective across all architecture variants. Fifteen carefully selected configurations were evaluated, representing combinations of the most influential parameters affecting contrastive learning performance.

The optimization focused on four critical parameters that directly impact the quality of learned representations in the contrastive learning framework. Temperature values of 0.05, 0.10, and 0.15 were evaluated to control the concentration of the contrastive distribution, with lower temperatures promoting harder negative mining and higher discrimination. Projection dimensions of 64, 128, 256, and 512 were tested to determine the optimal dimensionality for the contrastive projection head, balancing representational capacity with computational efficiency. Contrastive weight values of 0.8, 1.0, 1.2, and 2.0 were examined to determine the relative importance of contrastive loss in the joint training phase. Classification weight values of 0.5, 0.8, 1.0, and 1.5 were evaluated to balance the classification objective with the contrastive learning component.

Table 3 presents the comprehensive optimization results for the ViG-S-GELU architecture, with configurations ranked by combined validation performance score.

The optimal configuration identified through ViG-S-GELU optimization achieved 91.82% validation accuracy with the parameter combination of temperature = 0.05, projection_dim = 128, contrastive_weight = 0.8, and classification_weight = 1.0. This configuration was subsequently applied to all other ViG architectures to ensure consistent training conditions while leveraging optimization results from a representative mid-sized architecture that balances computational efficiency with model expressiveness. The low temperature value shows that harder negative mining was beneficial for this medical application, while the moderate projection dimension suggests that excessive representational capacity may lead to overfitting in the contrastive space.

## 4. Results and Analysis

### 4.1. Architecture Performance Comparison

Based on the optimal hyperparameter configuration identified through comprehensive optimization, we systematically evaluated all four ViG-GELU architecture variants to demonstrate the scalability and effectiveness of graph-based spectro-temporal analysis across different model complexities. To demonstrate the specific contribution of supervised contrastive learning to ViG performance, we conducted controlled experiments training each architecture with and without the contrastive learning component, enabling direct assessment of the enhancement provided by the contrastive learning framework.

The Table 4 results demonstrate substantial and consistent improvements across all ViG architectures when supervised contrastive learning is integrated. ViG-Ti-GELU showed the most dramatic improvement, with test accuracy increasing from 78.68% to 91.63%, representing a substantial gain of 12.95%. This improvement is particularly notable given the smallest model size, demonstrating exceptional sensitivity gains from 79.25% to 95.36%. This substantial improvement makes ViG-Ti particularly suitable for screening applications where computational efficiency and high sensitivity are crucial for clinical deployment.

ViG-S-GELU achieved significant enhancement from 82.64% to 89.99% test accuracy, representing a 7.35% improvement with balanced improvements across precision from 83.13% to 90.84% and recall from 82.16% to 92.01%. The AUC-ROC improvement from 0.886 to 0.961 shows substantially better discriminative capability, making this variant optimal for standard clinical deployment scenarios where balanced performance across metrics is essential.

ViG-M-GELU demonstrated robust performance gains from 85.22% to 91.78% test accuracy, achieving a 6.56% improvement and the highest overall performance metrics with 91.32% precision, 94.85% recall, and 0.966 AUC-ROC. The balanced excellence across all metrics positions ViG-M as the optimal choice for specialized diagnostic applications requiring maximum accuracy and reliability. This architecture represents the sweet spot between computational efficiency and diagnostic performance.

ViG-B-GELU showed consistent but more modest improvements from 84.81% to 89.24% test accuracy, representing a 4.43% gain. This suggests that the largest architecture may be approaching performance saturation or experiencing diminishing returns from increased complexity, possibly due to overfitting on the relatively limited medical dataset or the inherent limitations of the task complexity.

### 4.2. Confusion Matrix Analysis

The confusion matrices reveal comprehensive diagnostic performance across all ViG architectures, demonstrating the consistent improvements achieved through contrastive learning across all model scales. ViG-M-GELU with contrastive learning achieves optimal diagnostic performance with the highest true positive rate of 368 correct PD classifications and true negative rate of 246 correct healthy classifications, while maintaining the lowest combined error rate of 55 total errors comprising 35 false positives and 20 false negatives.

ViG-Ti-GELU with contrastive learning shows the highest sensitivity with 370 true positives, making it particularly suitable for screening applications where maximizing detection of positive cases is prioritized over minimizing false positives. This high sensitivity is crucial in medical screening contexts where missing a positive case has more severe consequences than generating false alarms that can be resolved through follow-up testing.

The comparison clearly illustrates in Figure 4 that supervised contrastive learning consistently reduces misclassifications across all architectures. ViG-Ti achieves 56 total errors compared to significantly higher error rates in non-contrastive variants, ViG-S achieves 67 total errors, ViG-M maintains 55 total errors, representing the best overall performance, and ViG-B produces 72 total errors. These results demonstrate that the contrastive learning framework provides consistent benefits regardless of model complexity, with the improvements being particularly pronounced in smaller architectures.

The error patterns across architectures reveal interesting insights into the nature of PD speech classification. False positive rates remain relatively low across all enhanced models, suggesting that the learned representations effectively capture the characteristics of healthy speech. False negative rates show more variation, with ViG-M achieving the optimal balance between sensitivity and specificity, making it most suitable for clinical diagnostic applications where both metrics are critical.

### 4.3. Feature Representation Quality Analysis

To evaluate the quality of learned representations, we performed t-SNE visualization analysis on embeddings extracted from different ViG-GELU architectures, as shown in Figure 5. This analysis quantifies how well the learned features separate PD-affected from healthy speech patterns in the high-dimensional representation space, which directly correlates with classification performance and clinical reliability achieved by the proposed approach.

Table 5 quantifies feature separability metrics across different ViG architectures using established clustering evaluation measures.

ViG-M-GELU demonstrates superior feature representation quality across all clustering metrics, establishing it as the optimal architecture for this medical application. The silhouette score of 0.5505 shows well-separated, cohesive clusters with minimal ambiguity between PD-affected and healthy speech representations, placing the approach in the “good clustering structure” range which is exceptional for medical data where subtle PD-affected speech differences are often challenging to capture.

The Calinski–Harabasz index of 1217.87 represents excellent cluster separation, indicating that learned features capture distinct PD speech characteristics that are clearly distinguishable from healthy controls and robust to recording artifacts and inter-patient variability. This high value suggests that the learned representations effectively encode the dysarthric speech patterns while maintaining consistency within diagnostic categories.

The optimal Davies–Bouldin index of 0.5845 shows compact, well-separated clusters that minimize diagnostic uncertainty and enable detection of subtle PD-related speech changes in early-stage disease. This metric is particularly important for clinical applications where minimizing false positives and false negatives is crucial for patient care and treatment decisions.

The exceptional separation ratio of 2.53 demonstrates that PD and healthy samples are 2.53 times farther apart between classes than within classes, providing a substantial margin for confident diagnostic predictions crucial for medical applications. This separation provides confidence that the learned representations capture clinically meaningful differences rather than spurious correlations or dataset-specific artifacts.

## 5. Discussion

The results demonstrated substantial advances in automated PD detection from speech signals through the integration of ViGs with supervised contrastive learning. Performance gains ranging from 4.43% to 12.95% in test accuracy across all architectural variants highlight the effectiveness of this approach compared to standard supervised training.

The superior performance of ViG architectures stems from their explicit modeling of relational structures within spectro-temporal representations. Unlike conventional CNNs that treat spectrograms as regular images, ViGs capture long-range dependencies between frequency bands and temporal segments crucial for detecting subtle PD-specific speech characteristics. The feature separability analysis confirms this advantage, with ViG-M-GELU achieving a separation ratio of 2.53, indicating that PD and healthy speech patterns are well-separated in the learned representation space.

The frequency band decomposition strategy effectively captures distinct disease manifestations across the spectrum. Low-frequency bands (0–2 kHz) reveal voice tremor and fundamental frequency instabilities, mid-frequency bands (2–6 kHz) capture formant modifications and articulation deficits, while high-frequency bands (6 kHz+) provide information about breathiness and noise characteristics often overlooked in traditional approaches.

The supervised contrastive learning framework addresses the fundamental challenge of limited labeled medical data while maintaining diagnostic relevance. The most dramatic improvement occurred in ViG-Ti-GELU, where test accuracy increased from 78.68% to 91.63% (12.95% gain), suggesting that contrastive learning provides strong regularization effects that prevent overfitting while enhancing discriminative feature learning from limited examples.

Cross-institutional validation across Italian, Colombian, and Spanish datasets represents a significant methodological strength that addresses fundamental limitations in current PD detection research. Consistent performance across different languages, recording environments, and demographic populations suggests that learned representations capture universal PD-specific speech characteristics rather than dataset-specific artifacts. This finding aligns with recent studies demonstrating that smartphone-recorded voice samples can provide reliable PD detection capabilities across diverse populations [12,14], supporting the clinical viability of our approach.

While our cross-institutional validation provides unprecedented diversity, several methodological considerations require acknowledgment. Recording equipment differences across institutions were addressed through comprehensive technical harmonization procedures, including resampling, dynamic range normalization, and consistent preprocessing pipelines. Potential cultural and linguistic variations in speech production patterns were minimized through strategic focus on sustained vowels, which exhibit greater cross-linguistic consistency than connected speech across Romance languages. Different clinical assessment protocols across institutions, rather than representing limitations, reflect real-world clinical diversity that may actually enhance model robustness for practical deployment across varied healthcare environments.

The exclusive use of sustained vowel phonations, while limiting generalizability to connected speech, provides several methodological advantages essential for this cross-institutional study. Controlled phonetic content enables reliable cross-linguistic comparison, standardized elicitation protocols maintain consistency across institutions, and sustained vowels provide optimal conditions for our frequency band decomposition approach. However, we acknowledge that this limitation constrains the ecological validity of our findings, as connected and spontaneous speech tasks would better capture the full spectrum of PD-related speech impairments encountered in real-world clinical scenarios.

Connected speech analysis would enable detection of additional PD manifestations, including prosodic alterations, articulatory coordination deficits, fluency disruptions, and speech timing irregularities that are not captured in sustained phonations. These aspects are clinically significant as they directly impact communication effectiveness and quality of life in PD patients. Furthermore, spontaneous speech tasks would provide more naturalistic assessment conditions that better reflect patients’ everyday communication challenges, potentially offering enhanced sensitivity to early-stage disease manifestations.

Despite these limitations, sustained vowels offer unique advantages for cross-institutional research. They minimize linguistic variability that could confound cross-cultural comparisons, enable precise spectral analysis of voice quality changes fundamental to PD pathophysiology, and provide standardized elicitation protocols suitable for diverse clinical environments. The voice quality changes captured through sustained vowels—including fundamental frequency instability, voice tremor, and breathiness—represent core laryngeal manifestations of PD that are present across all speech tasks and serve as reliable early indicators of disease progression.

This cross-linguistic generalization is crucial for clinical deployment, indicating potential adaptation to different languages without extensive retraining. Future work will include a comprehensive investigation of connected speech tasks, including reading passages, picture descriptions, and spontaneous monologues, to capture broader aspects of PD speech impairment. Additionally, we plan to conduct cross-dataset performance evaluation (training on individual datasets and testing across institutions) to further validate generalizability and explore the integration of multiple speech tasks for enhanced diagnostic accuracy and broader clinical applicability.

The achieved test accuracy of 91.78% with ViG-M-GELU represents competitive performance within current speech-based PD detection systems. The approach demonstrates superior generalization compared to methods reporting high accuracies on single-institution datasets but failing to maintain performance across different populations. The integration of graph neural networks with medical speech analysis represents a novel contribution, demonstrating that treating spectrograms as graphs enables more effective modeling of complex relationships in PD speech patterns.

Clinically, the high sensitivity of ViG-Ti-GELU (95.36%) makes it suitable for screening applications, while the balanced performance of ViG-M-GELU (91.32% precision, 94.85% recall) positions it optimally for diagnostic applications. These performance levels are competitive with other recent screening approaches, such as wavelet scattering networks, which have shown similar promise for early PD detection [11]. The speech-based approach offers significant deployment advantages, enabling remote assessment and longitudinal monitoring using standard devices without specialized equipment or trained personnel.

Additional limitations beyond the cross-institutional considerations require acknowledgment. The dataset, while representing three institutions, remains limited compared to large-scale medical studies and would benefit from expansion to additional languages and clinical populations. Future work should investigate connected speech tasks to capture broader aspects of PD speech impairment, larger diverse populations for enhanced generalizability, disease severity modeling for longitudinal assessment, and extension to other neurological conditions affecting speech production. The integration of smartphone-based data collection and real-world deployment validation represents important next steps for clinical translation.

## 6. Conclusions

This study successfully integrates ViGs with supervised contrastive learning for enhanced PD detection from speech signals. The ViG-M-GELU architecture achieved optimal performance with 91.78% test accuracy, 91.32% precision, 94.85% recall, and 0.966 AUC-ROC, representing substantial improvements over baseline approaches. Consistent improvements across all architectural variants demonstrate framework effectiveness, with particularly dramatic gains in the smallest architecture benefiting resource-constrained applications.

The approach addresses fundamental limitations through graph-based modeling of spectro-temporal relationships and supervised contrastive learning that effectively leverages limited labeled medical data. Cross-linguistic validation across Italian, Colombian, and Spanish datasets provides evidence for broader clinical applicability. The framework offers practical deployment advantages through speech-based remote assessment capabilities using standard devices.

This work demonstrates graph neural network effectiveness for medical signal analysis and establishes a foundation for developing accurate, accessible diagnostic tools for PD. The framework can potentially extend to other neurological conditions and multimodal assessment approaches, contributing to the growing evidence supporting artificial intelligence in transforming neurological disease detection for earlier intervention and improved patient outcomes.

## Figures and Tables

**Figure 1 jimaging-11-00220-f001:**
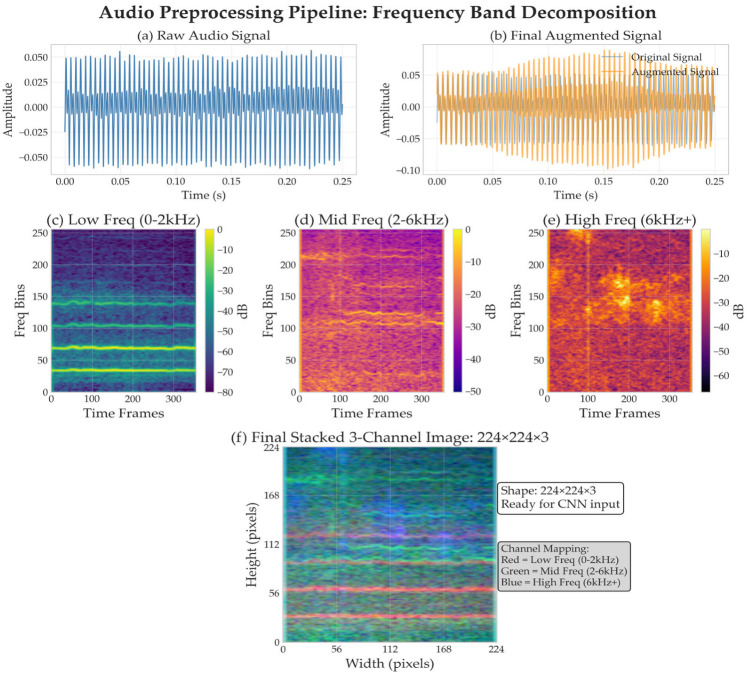
Audio preprocessing pipeline: frequency band decomposition. (**a**) Raw audio signals (**b**) augmented and, (**c**–**e**) decomposed into three frequency bands: low (0–2 kHz), mid (2–6 kHz), and high (6 kHz+) spectrograms, and (**f**) stacked into a 224 × 224 × 3 RGB-like tensor for neural network input.

**Figure 2 jimaging-11-00220-f002:**
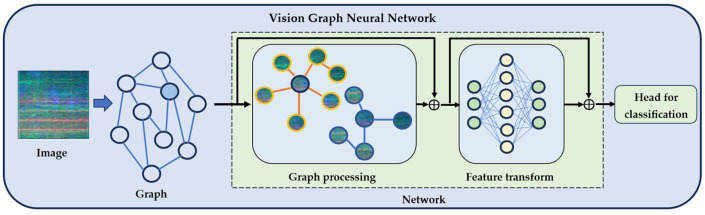
Vision graph neural network architecture for audio classification.

**Figure 3 jimaging-11-00220-f003:**
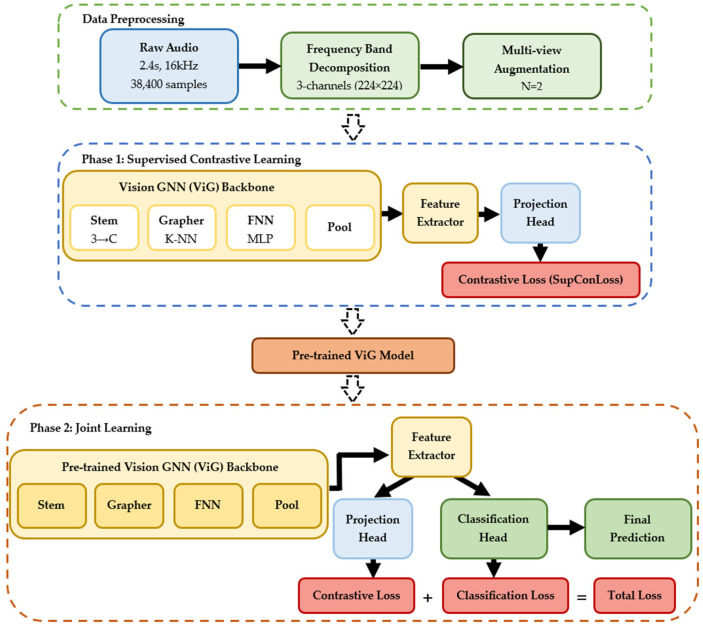
Two-phase training architecture. Overview of the proposed methodology: Data preprocessing transforms raw audio into 3-channel frequency band spectrograms with multi-view augmentation.

**Figure 4 jimaging-11-00220-f004:**
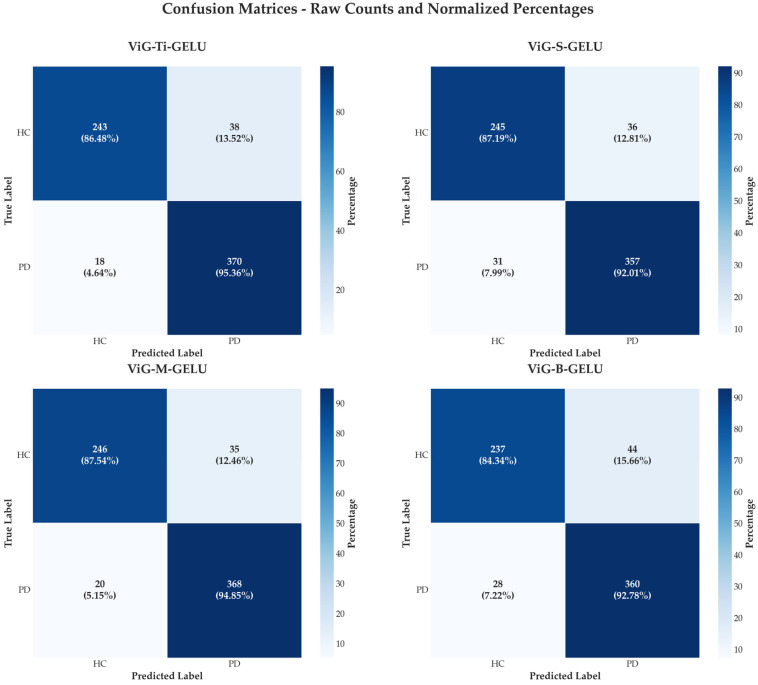
Confusion matrix analysis for ViG architectures with supervised contrastive learning on PD voice classification. Confusion matrices showing classification performance of four ViG architectures (Ti, S, M, B) with GELU activation and supervised contrastive learning on PD voice data. Key takeaways: (1) ViG-M-GELU achieves optimal diagnostic balance with lowest combined error rate (55 total errors), (2) ViG-Ti-GELU demonstrates highest sensitivity (370 true positives) making it suitable for screening applications, (3) Error trends show consistently low false positive rates across all models, indicating effective healthy speech characterization, (4) The systematic error reduction pattern (Ti: 56, S: 67, M: 55, B: 72 total errors) demonstrates contrastive learning effectiveness regardless of model complexity.

**Figure 5 jimaging-11-00220-f005:**
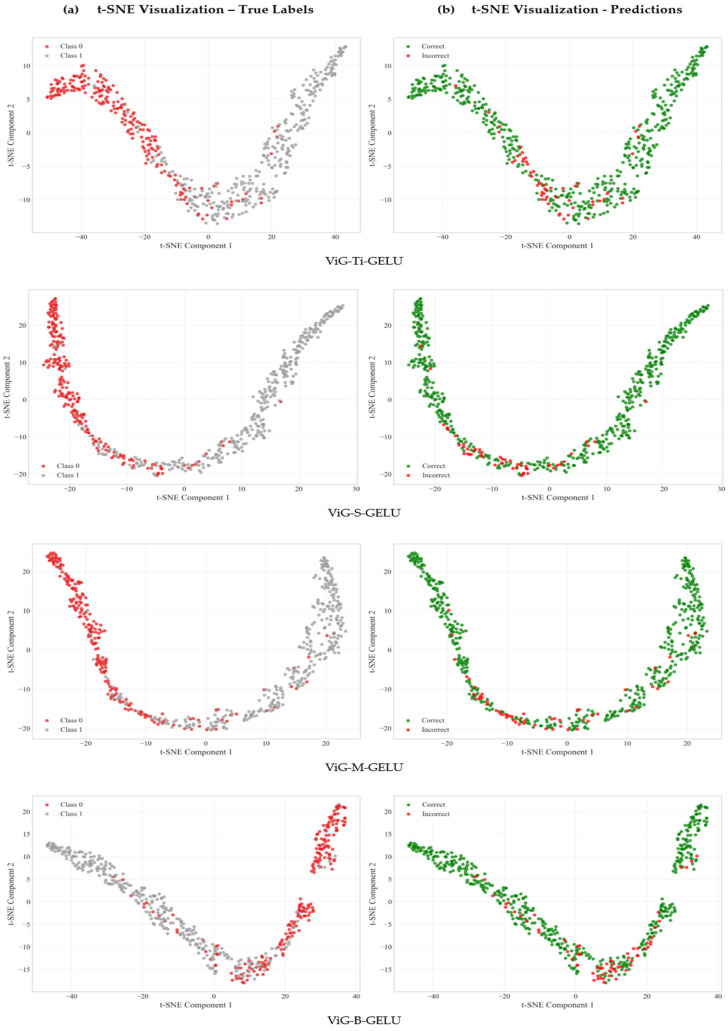
t-SNE feature representation analysis description: Multi-panel visualization showing t-SNE projections of learned features from all ViG architectures. Each panel displays two sub-plots: (**a**) true class labels (Class 0: Healthy Control, Class 1: PD) and (**b**) prediction accuracy (Correct: green, Incorrect: red). Key findings: ViG-M-GELU demonstrates the clearest class separation with minimal overlap, concentrated error regions indicate systematic misclassifications, and well-defined clustering confirms that learned representations capture clinically meaningful differences between PD-affected and healthy speech patterns.

**Table 1 jimaging-11-00220-t001:** Multi-Institutional dataset characteristics.

Dataset	Participants	Demographics (Age)	Country/ Institution	Complete Speech Tasks Available	Audio Specs
Italian PVS	50 (28 PD, 22 HC)	67.1 ± 5.2 (HC), 67.2 ± 4.8 (PD)	Italy/University of Bari	Sustained vowels (/a/, /e/, /i/, /o/, /u/), phonemically balanced text reading, syllable repetitions (/pa/, /ta/), phonemically balanced words	16 kHz (harmonized)
PC-GITA	100 (50 PD, 50 HC)	31–86 years, gender-matched	Colombia/ Clínica Noel, Medellín	Sustained vowels (/a/, /e/, /i/, /o/, /u/), diadochokinetic evaluation, 45 isolated words, 10 sentences, reading text, monologues	44.1 kHz, 16-bit
Neurovoz	108 (53 PD, 55 HC)	Age-matched adults	Spain/UPM and Hospital Gregorio Marañón	Sustained vowels (/a/, /e/, /i/, /o/, /u/) in triplicate, diadochokinetic tests (/pa-ta-ka/), 16 listen-and-repeat utterances, spontaneous monologues	16 kHz
Combined (This Study)	258 (131 PD, 127 HC)	Multi-demographic	Multi institutional	Sustained vowels (/a/, /e/, /i/, /o/, /u/) only	16 kHz (harmonized)

**Table 2 jimaging-11-00220-t002:** Final dataset distribution after preprocessing and segmentation.

Split	Total Segments	Healthy Control	PD	Class Balance Ratio (HC: PD)	Augmentation Applied
Training	6027	2616 (43.35%)	3411 (56.65%)	1:1.30	Multi-view + Augment
Validation	660	324 (49.10%)	336 (50.90%)	1:1.04	No
Test	669	321 (48.00%)	348 (52.00%)	1:1.08	No
Total	7356	3261 (43.88%)	2103 (56.12%)	1:1.26	-

**Table 3 jimaging-11-00220-t003:** Supervised contrastive learning hyperparameter optimization results.

Rank	Val Accuracy	Val Loss	Temperature	Projection Dimensions	Contrastive Weight	Classification Weight
1	91.82%	1.247	0.05	128	0.8	1.0
2	90.60%	1.428	0.05	128	1.2	0.8
3	90.15%	1.568	0.05	256	1.2	1.0
4	89.54%	1.259	0.05	256	1.0	1.0
5	89.39%	1.921	0.10	128	1.0	1.0
6	88.93%	2.560	0.10	256	1.2	1.0
7	88.18%	2.177	0.10	256	1.0	1.5
8	87.72%	1.933	0.10	64	1.0	1.0

**Table 4 jimaging-11-00220-t004:** ViG-GELU architecture performance comparison with and without supervised contrastive learning.

Architecture	Training Acc	Validation Acc	Test Acc	Precision	Recall	F1-Score	AUC-ROC
ViG-Ti-GELU + Contrastive Learning	98.46%	89.85%	91.63%	90.69%	95.36%	92.96%	0.965
ViG-Ti-GELU	79.38%	79.94%	78.68%	79.25%	78.68%	79.25%	0.851
ViG-S-GELU + Contrastive Learning	99.24%	89.24%	89.99%	90.84%	92.01%	91.42%	0.961
ViG-S-GELU	83.16%	81.83%	82.64%	83.13%	82.16%	82.62%	0.886
ViG-M-GELU + Contrastive Learning	99.26%	90.52%	**91.78%**	**91.32%**	**94.85%**	**93.05%**	**0.966**
ViG-M-GELU	85.28%	84.77%	85.22%	85.64%	84.82%	85.26%	0.911
ViG-B-GELU + Contrastive Learning	99.86%	88.67%	89.24%	89.11%	92.78%	90.91%	0.950
ViG-B-GELU	84.82%	84.26%	84.81%	85.28%	84.46%	84.83%	0.907

**Table 5 jimaging-11-00220-t005:** Feature separability analysis across ViG-GELU architectures.

Architecture	Silhouette Score	Calinski–Harabasz Index	Davies-Bouldin Index	Intra-Class Distance	Inter-Class Distance	Separation Ratio
ViG-Ti-GELU	0.4788	938.92	0.6712	22.21	46.83	2.11
ViG-S-GELU	0.4840	914.15	0.6384	38.78	81.02	2.09
ViG-M-GELU	0.5505	1217.87	0.5845	30.13	76.29	2.53
ViG-B-GELU	0.4227	739.03	0.7829	37.48	70.64	1.88

## Data Availability

The datasets used in this study are publicly available or accessible upon request: Italian PVS corpus via IEEE DataPort (https://ieee-dataport.org/open-access/italian-parkinsons-voice-and-speech, accessed on 29 June 2025), PC-GITA corpus through Universidad de Antioquia upon academic request, and Neurovoz corpus through Universidad Politécnica de Madrid following established data sharing agreements visit: https://zenodo.org/records/10777657, accessed on 29 June 2025. All datasets were used in accordance with their respective usage terms and ethical guidelines.

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
