# Peer review of "Spectro-Image Analysis with Vision Graph Neural Networks and Contrastive Learning for Parkinson’s Disease Detection"

_2313-433X, 2025, doi:10.3390/jimaging11070220_

Round 1

Reviewer 1 Report

Comments and Suggestions for Authors

The manuscript presents a method for Parkinson’s Disease detection by applying ViG on Images obtained   from Image data. The method presented appears to achieve high classification performance, however there are few comments which require attention before this work can be recommended for publication.

  • The introduction lacks clear motivation behind choosing this architecture. The literature review presented does not look sufficient to lead to this model.
  • The research question appears vague. It is not clear why vision-based model was adopted in the first place. Although the introduction introduces some prior works using such methods, it is still not sufficiently stressed.
  • Regarding the use of dataset, the manuscript does not provide any qualitative and quantitative analysis related to combining the datasets. Although the goal appears to be introducing diversity of contexts, these datasets differ in various ways. It is advisable to justify such use through appropriate analysis.
  • Line 132, It’s not clear what “Romance language” means in this sentence.
  • Line 318, Which dataset was the backbone ViG model pre-tained on. As the pre-training might have affected the results, it is worth-mentioning.

Overall, the presented method despite having improved classification performance, the experiment design needs more explanation as suggested in the comments.

Author Response

Comments 1: The introduction lacks clear motivation behind choosing this architecture. The literature review presented does not look sufficient to lead to this model.

Response 1: Thank you for this important observation. We have significantly enhanced the introduction section to provide clear motivation for selecting Vision Graph Neural Networks. [Revision: Page 3, Lines 105-116] We have added the following explanation:

“The selection of Vision Graph Neural Networks for this application is motivated by three key limitations in current approaches: First, traditional CNNs process spectrograms as regular images, failing to capture the complex relational dependencies between frequency bands and temporal segments that are crucial for detecting subtle PD-specific speech characteristics. Second, the spatial relationships in spectro-temporal data are inherently graph-like, where frequency bins and time frames exhibit long-range dependencies that standard convolution operations cannot effectively model. Third, the adoption of vision-based models for speech analysis represents a paradigm shift that treats audio signals as spectro-images, enabling the application of advanced computer vision techniques to capture dysarthric speech patterns that manifest as complex spectro-temporal relationships across different frequency bands, which are better modeled through graph structures than conventional grid-based representations.”

Comments 2: The research question appears vague. It is not clear why vision-based model was adopted in the first place. Although the introduction introduces some prior works using such methods, it is still not sufficiently stressed.

Response 2: We have reformulated our research question to be more specific and clearly articulated. [Revision: Page 3, Lines 117-122] The updated research question now reads:

“This study addresses the fundamental research question: Can Vision Graph Neural Networks combined with supervised contrastive learning effectively model the complex spectro-temporal relationships in speech signals to achieve superior Parkinson's Disease detection compared to traditional CNN-based approaches? Specifically, we investigate whether explicitly modeling spectrograms as graphs rather than regular images enables better capture of the subtle, distributed patterns characteristic of dysarthric speech.”

Comments 3: Regarding the use of dataset, the manuscript does not provide any qualitative and quantitative analysis related to combining the datasets. Although the goal appears to be introducing diversity of contexts, these datasets differ in various ways. It is advisable to justify such use through appropriate analysis.

Response 3: We acknowledge this important methodological concern and have added comprehensive justification for our multi-institutional approach. [Revision: New Section 2.1.3, Pages 5-6, Lines 206-232] We have added a complete new section "Cross-Institutional Dataset Integration and Validation" that explains the phonetic compatibility across Romance languages, demographic complementarity, and methodological advantages. Additionally, we have updated the audio preprocessing section [Revision: Page 6, Lines 240-247] to explain how recording heterogeneity was addressed through standardized preprocessing procedures.

Comments 4: Line 132, It's not clear what "Romance language" means in this sentence.

Response 4: We have clarified the meaning of "Romance language" [Revision: Page 4, Lines 149-151] by modifying the text to:

"The strategic combination provides unprecedented diversity in recording environments, equipment specifications, demographic distributions, and clinical protocols while maintaining phonetic consistency through sustained vowel productions (/a/, /e/, /i/, /o/, /u/) across Italian, Spanish, and Colombian Spanish—all belonging to the Romance language family, which share similar vowel systems and phonetic characteristics relevant for PD detection."

Comments 5: Line 318, Which dataset was the backbone ViG model pre-trained on. As the pre-training might have affected the results, it is worth-mentioning.

Response 5: This is an excellent point regarding the potential impact of pre-training on results. [Revision: Page 10, Lines 363-372] We have added detailed explanation in section 2.3.1 clarifying that we used random weight initialization following Xavier/Glorot initialization rather than traditional ImageNet pre-training, and employed a domain-specific two-stage training strategy to avoid potential domain mismatch between natural images and audio spectrograms.

Reviewer 2 Report

Comments and Suggestions for Authors
  1. While the authors provide a strong rationale for using sustained vowel phonations, this decision limits the generalizability of the findings. Connected or spontaneous speech better represents real-world scenarios. I suggest expanding the discussion in Section 5.

  2.  

    The authors used different augmentations (e.g., pitch shifting, time stretching), but it's unclear whether these transformations mimic PD-induced variations or just help regularize the model. Add references or clarification.

  3. The t-SNE and confusion matrices are informative. However, captions for these figures could be expanded to describe key takeaways (e.g., specific clustering benefits, error trends across classes).
  4.  

    It would help readers if the dataset links hyperlinked in the Data Availability Statement section.

Author Response

Comments 1: While the authors provide a strong rationale for using sustained vowel  phonations, this decision limits the generalizability of the findings. Connected or  spontaneous speech better represents real-world scenarios. I suggest expanding the  discussion in Section 5. 

Response 1: We appreciate this important observation regarding ecological validity.  [Revision: Pages 19-20, Lines 676-720] We have substantially expanded the discussion section to provide comprehensive rationale for using sustained vowel phonations, acknowledging  limitations while explaining methodological advantages. We have also outlined detailed  future work plans to extend to connected speech tasks including reading passages, picture descriptions, and spontaneous monologues. 

Comments 2: The authors used different augmentations (e.g., pitch shifting, time  stretching), but it's unclear whether these transformations mimic PD-induced variations or  just help regularize the model. Add references or clarification. 

Response 2: This is an excellent point regarding clinical validity of our augmentation strategy.  [Revision: Pages 8-9, Lines 309-338] We have substantially enhanced section 2.2.2 to provide  strong scientific rationale for each augmentation technique, grounding them in the known  pathophysiology of Parkinsonian speech disorders with appropriate references. Each  augmentation technique is now clearly justified as clinically relevant rather than arbitrary  regularization. 

Comments 3: The t-SNE and confusion matrices are informative. However, captions for  these figures could be expanded to describe key takeaways (e.g., specific clustering  benefits, error trends across classes). 

Response 3: [Revision: Figure 4 caption, Page 16, Lines 578-586; Figure 5 caption, Page 18,  Lines 636-642] We have enhanced both figure captions with detailed key takeaways, including  specific diagnostic insights, error trend analysis, and clinical implications of the visualization  results. 

Comments 4: It would help readers if the dataset links hyperlinked in the Data  Availability Statement section. 

Response 4: [Revision: Page 21, Lines 775-780] We have ensured proper formatting of dataset  links in the Data Availability Statement section and will implement clickable hyperlinks  during final formatting if journal format permits. 

Reviewer 3 Report

Comments and Suggestions for Authors

The authors have introduced a novel framework combining Vision Graph Neural Networks (ViG) with supervised contrastive learning for the detection of Parkinson’s Disease (PD) from speech signals. This integration is methodologically novel in the detection of neurological disorders such as PD and addresses key limitations in speech-based AI disease detection, such as the lack of large labeled datasets.

Overall, I recommend the manuscript to be accepted in its present form.

Strengths:

  • Excellent methodological rigor demonstrated with the use of diverse datasets, and detailed and well-established preprocessing routines including a robust data augmentation technique
  • The methods are commendable, replicable, and offer good performance comparisons.
  • Rigorous, well-rounded presentation of performance metrics such as accuracy, AUC, etc.
  • The generalizability and overall robustness of the underlying analyses make this technique a good candidate for PD detection in the real world.

Weaknesses:

  • As addressed, the analyses is limited to sustained vowel phonations; extending to extensive naturalistic speech data, perhaps in a future expansion of the study would improve the comprehensiveness of the study and stronger validation of the technique

Comments:

  • Line 123: Incorrect use of acronym: replace ‘Pd’ with ‘PD’

Author Response

Comments 1: Line 123: Incorrect use of acronym: replace ‘Pd’ with ‘PD’ 

Response 1: [Revision: Page 3, Line 123] We have corrected the acronym from "Pd" to "PD" 
as requested. 

We appreciate the reviewer's positive assessment of our methodological rigor, diverse 
datasets, and potential real-world applicability. We are pleased that our integration of Vision 
Graph Neural Networks with supervised contrastive learning was recognized as 
methodologically novel for neurological disorder detection.
